

**Technical note: Investigating saline water uptake by roots using**
**spectral induced polarization**
Solomon Ehosioke[1, 7*], Sarah Garré[2, 3], Johan Alexander Huisman[4], Egon Zimmermann[5], Mathieu Javaux[4, 6], and
Frédéric Nguyen[1]
[1]Urban and Environmental Engineering, Liège University, Liège, Belgium
[2]Flanders Research Institute for Agriculture, Fisheries and Food, Melle, Belgium
[3]Gembloux Agro-Bio Tech, Liège University, Gembloux, Belgium
[4]Agrosphere (IBG 3), Forschungszentrum Jülich GmbH, Jülich, Germany
[5]Electronic Systems (ZEA-2), Forschungszentrum Jülich GmbH, Germany
[6]Earth and Life Institute, Environmental Science, UCLouvain, Louvain-la-Neuve, Belgium
[7]Land, Air and Water Resources, University of California, Davis, USA
**Corresponding Author:** Solomon Ehosioke (sehosioke@ucdavis.edu; solomon.ehosioke@gmail.com)



**Abstract**

There has been some improvements in the methods available for root investigation in recent years that has enabled many studies to be carried out on the root, which represents the hidden half of the plant. Despite the increased studies on roots, there are still knowledge gaps in our understanding of the electromagnetic processes in plant roots which will be useful to quantify plant properties, and monitor plant physiological responses to dynamic environmental factors amidst climate change. In this study, we evaluated the suitability of spectral induced polarization for non-invasive assessment of root activity. We investigated the electrical properties of the primary roots of Brachypodium distachyon L. and Zea mays L. during the uptake of fresh and saline water using SIP measurements in a frequency range from 1 Hz to 45 kHz. The results show that SIP is able to detect the uptake of water and saline water in both species, and that their electrical signature were influenced by the solute concentration. The resistivity and phase response of both species increased with solute concentration until a certain threshold before it decreased. This concentration threshold was much higher in Maize than in Brachypodium, which implies that tolerance to salinity varies with the species, and that Maize is more tolerant to salinity than Brachypodium. We conclude that spectral induced polarization is a useful tool for monitoring root activity, and could be adapted for early detection of salt stress in plants.

**Keywords:** Agrogeophysics, Spectral induced polarization, Salt stress, Maize roots, Brachypodium roots





### 1. Introduction

Sustainable global crop production is challenged by several unfavorable envirnmental factors such as drought, extreme temperatures, salinity, nutrient deficiency, and soil contamination among others. For example, more than 800 million ha of land globally is affected by salinity and excessive sodium content (FAO 2005; Munns 2005). High salt concentrations in soils induce plant stress due to low external water potential, ion toxicity ($Na^+$ and/or $Cl^-$) or nutrient deficiency by interfering with the uptake and transport of various essential nutrients (Munns et al. 2006; Läuchli and Grattan 2012; Hussain et al. 2013; Negrao et al. 2017; Isayenkov and Maathius 2019). Stress magnitude depends on the species, duration of salinity exposure, the growth stage and environmental conditions (Munns and Tester 2008). Accumulation of sodium and chloride ions at toxic levels in plant tissue damages biological membranes and subcellular organelles, reducing plant growth and development (Davenport et al. 2005; Zhao et al. 2010; Farooq et al. 2015; Isayenkov and Maathuis 2019). Sodium may also displace calcium from the binding site of the cell membrane which can result in membrane leakiness (Cramer et al. 1988). Geophysical electrical methods have extensively been used to study root water uptake in soils (e.g. Michot et al. 2003; Garré et al. 2011; Beff et al. 2013) and soil salinity (e.g. Rhoades et al. 1999; Bennett et al. 2000; Doolittle et al. 2001; Ben Hamed et al. 2016; Shahnazaryan et al. 2018). Due to their sensitivity to salinity, they provide a natural means to non-invasively study salt stress impact on roots given the analogy between water flow and electrical current flow in roots. Spectral induced polarization (SIP), also known as electrical impedance spectroscopy (EIS), has been successfully used to study various plant physiological processes, such as growth (Ozier-Lafontaine and Bajazet 2005; Repo et al. 2005), mycorrhizal colonization (Cseresnyés et al. 2013; Repo et al. 2014), cold acclimation (Repo et al. 2016), nutrient deprivation (Weigand and Kemna 2017, 2019), and the effects of salt stress on growth (Ben Hamed et al. 2016). In the interpretation of these SIP measurements, it is assumed that current pathways in the extracellular (apoplast) and intercellular (plasmodesmata) spaces play an important role in electrical charge migration and storage (Kinraide, 2001; Kinraide and Wang, 2010, Weigand and Kemna, 2019; Kessouri et al., 2019) (Fig. 1). In particular, current conduction is assumed to depend on the electrical properties of the apoplast and the ionic composition of the extracellular fluid (ECF), whereas polarization is assumed to occur at the cell membrane interface because charged particles such as $Na^+$, $Ca^{2+}$, $K^+$, $Cl^-$ ions and amino acids cannot diffuse directly across the cell membrane. Instead, they can only cross the membrane through ion pumps and ion channels, whose opening and closing are regulated by the membrane potential difference. Polarization is also expected to occur at the outer root surface (i.e. the root-soil interface), where the charge distribution that determines polarization depends on the concentration of ions in the external fluid (Weigand and Kemna 2017, 2019).





Conduction and polarization mechanisms are frequency dependent (see current pathways in Fig. 1b and 1c) and
can be assessed simultaneously by measuring the frequency dependent electrical impedance of a biological tissue
using SIP. The suitability of this method for investigating root responses to salt stress is not well known and has
rarely been studied (Ben Hamed et al. 2016).

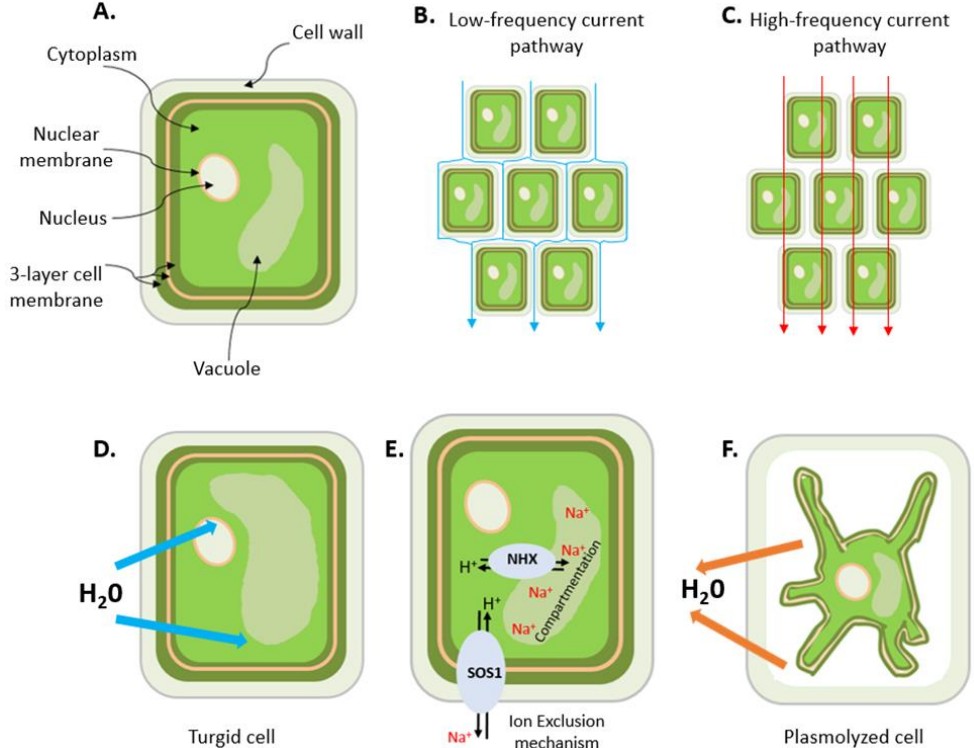


**Figure 1.**  Schematic illustration of **a**) plant cell showing some of the organelles (vacuole, nucleus and nuclear membranes),
the cell wall and the 3-layer (protein-lipid-protein) cell membrane, **b**) low-frequency current pathway, **c**) high frequency current
pathway, **d**) turgid cell resulting from the uptake of water, **e**) early stage response to salt stress in a plant root cell (adapted from
Deinlein et al. 2014), this involves the activation of cellular detoxification mechanisms, including NHX and SOS Na$^+$ transport
mechanisms (NHX: Na$^+$/H$^+$ exchanger, SOS: Salt Overly Sensitive), **f**) plasmolyzed cell due to excessive loss of water. This
can occur at a later stage of salt stress, when there are excess ions in the solution because the root cells can no longer exclude
or compartment them into the vacuole, water leaves the cell by osmosis leading to plasmolysis.

Plants respond to salt stress by adaptive mechanisms such as root exclusion of excess sodium in the surrounding
water or compartmentation, removing toxic ions from the cytoplasm where sensitive metabolic processes occur
(Hasegawa et al. 2000; Munns and Tester 2008; Zhao et al. 2020) into the vacuole (Neubert et al. 2005; Farooq et



al. 2015; Isayenkov and Maathuis 2019). These two adaptive mechanisms are independent but their effectiveness
varies across species (Grieve et al. 2012; Acosta-Motos et al. 2017). They modify the ionic composition of the
extracellular and intracellular fluids (Fig. 1e), which suggests that these adaptive mechanisms can possibly also be
detected by SIP. For example, Ben Hamed et al. (2016) investigated the use of EIS to non-invasively assess salt
resistance and the signaling and short-term (0-240 minutes) response of Sea rocket (*Cakile maritima*) to salinity.
Sea rocket was used as a model for salt-tolerant plants as it can survive extended contact with solute concentrations
up to 500 mM NaCl. It accumulates salt ions preferentially in its leaves without dehydration and nutritional
disorders (Debez et al. 2013). Ben Hamed et al. (2016) found that the frequency-dependent impedance of leaves
changed with increasing salinity as well as the duration of stress for plants grown in sand and hydroponic culture
conditions. In particular, it was observed that for a group of 10 plants exposed to increasing salinity, the electrical
resistance of the leaves increased in the presence of 50-100 mM NaCl, but decreased for salinity above 100 mM
NaCl, with the lowest value observed at 400 mM NaCl. For another group of 10 plants exposed to a 400mM NaCl
treatment over 240 minutes, the electrical resistance increased at early stages of salt stress and reached a maximum
after 180 minutes before declining rapidly. They concluded that the increasing electrical resistance within the
tolerable range of salinity for growth (50–100 mM NaCl) indicated low salt movement in leaf cells due to
compartmentation of salt ions in the leaf vacuoles, as reported in previous studies (e.g. Debez et al. 2004; Ellouzi
et al. 2011). The decrease in electrical resistance at salinities above 100 mM NaCl was interpreted as an indication
of increased movement of salt ions in the leaf cells, most probably in the apoplastic space. They suggested that at
these higher salinities, leaf cells seemed to lose their ability to compartment all salt ions in the vacuoles. Therefore,
ions may have accumulated in the apoplast and caused osmotic and nutritional imbalances that led to stunted
growth. Similarly, Ellouzi et al. (2011) reported rapid accumulation of $Na^+$ in the vacuole and re-establishment of
osmotic homeostasis shortly after salt treatment (400 mM NaCl for 4 h). They also observed a decrease in the
electrical resistance of leaves of salt-treated plants, which was closely correlated with the increased accumulation
of $Na^+$ in the vacuole. These studies suggest that the electrical resistance of salt-stressed plants varies with degree
of salinity and the duration of salt stress. This implies that that the accumulation of $Na^+$ and $Cl^-$ ions in the
cytoplasm and apoplast will take a long time to reach toxic levels when the salt concentration is low.  At very high
salt concentrations, it is expected that toxic level will be attained much faster, this could happen in a couple of
minutes (e.g. Ben Hamed et al. 2016).
Despite these interesting studies, the suitability of SIP as a tool to study plant response to salinity has not been
thoroughly investigated and few existing studies focused mainly on plant leaves. More studies are still needed to



better understand how roots respond to salt stress. Therefore, the aim of this study is to evaluate the SIP response
of *Brachypodium* and *Maize* primary roots subjected to different levels of salinity and to link the observed changes
in electrical properties with the salt adaptation mechanisms of plants to obtain further insights into the ability of
SIP to detect salt stress in plant roots.
**2. Materials and methods**

**2.1. Investigated plants and salt solutions**
Brachypodium (Brachypodium distachyon L.) and Maize (Zea mays L.) were studied under different salinity
treatments. Brachypodium distachyon L. is a salt-sensitive plant that can tolerate salt stress below 200 mM NaCl
(e.g. Lv et al. 2014; Guo et al. 2020). Zea mays L. is moderately sensitive to salt stress (Kaddah and Ghowail
1964; Farooq et al. 2015) and can tolerate relatively high salinity up to 400 mM NaCl (e.g. de Azevedo Neto et al.
2004), depending on the genotype. Plants of both species were grown in the laboratory under daylight conditions
(without artificial light), normal humidity and an average temperature of 23.2°C. They were grown in plastic tubes
(5 x 20 cm) using a mixture of fine and coarse sand with a grain size distribution ranging from 0.1 to 1.0 mm
(Ehosioke et al. 2023). The plants were watered with tap water at 2-day intervals and were sampled at 20 days
after sowing (DAS). The average diameter of the Brachypodium and Maize primary roots were 0.22 mm and 0.89
mm, respectively. Both plant types were in the 3-leaves stage at the time of measurement. Before each SIP
measurement, the plant was removed from the growth tube and the sand particles on the roots were removed gently.
Salt solutions were prepared by dissolving sodium chloride (NaCl) in demineralized water. The electrical
conductivity was measured using a conductivity meter (HQ14D, HACH, Mechelen, Belgium). A total of 14 salt
solutions with different concentrations were prepared (Table 1). The resulting concentration is presented in ppm.
The nomenclature to describe different types of saline water based on concentration and electrical conductivity is
presented in Table A1 (see Appendix).
**Table 1** Description of salt solutions used during the experiments.

| Salt solution: mass of NaCl dissolved in 0.05 L of demineralized water (mg) | Concentration (ppm) | Concentration (mM) | Conductivity (mS/cm) | Temperature (°C) |
|---|---|---|---|---|
| Demineralized water (baseline) | - | - | 0.0012 | 24.8 |
| 50 | 1000 | 17.1 | 1.94 | 22.9 |




| 100 | 2000 | 34.2 | 3.20 | 22.6 |
|---|---|---|---|---|
| 150 | 3000 | 51.3 | 5.46 | 22.6 |
| 200 | 4000 | 68.4 | 6.78 | 22.5 |
| 300 | 6000 | 102.7 | 9.75 | 22.6 |
| 400 | 8000 | 136.9 | 12.66 | 22.7 |
| 500 | 10000 | 171.1 | 15.47 | 22.6 |
| 840 (Salt-L) | 16800 | 287.5 | 28.50 | 24.8 |
| 1690 | 33800 | 578.4 | 47.40 | 23.6 |
| 1700 | 34000 | 581.8 | 48.70 | 23.6 |
| 1750 | 35000 | 598.9 | 50.10 | 23.5 |
| 1800 | 36000 | 616 | 51.60 | 23.5 |
| 2000 | 40000 | 684.5 | 57.30 | 23.4 |
| 3000 (Salt-H) | 60000 | 1,026.7 | 83.40 | 25.3 |


## 2.2. Measurement set-up

The measurement set-up consists of a precision balance (Mettler PM 2000), sampling container, SIP measurement
system, and a sample holder especially designed for root segments (Fig. 2; Ehosioke et al. 2023). We used the high
precision balance for a precise measurement of the uptake. The SIP measurement system is made up of a data
acquisition (DAQ) card (NI USB-4431), an amplifier unit (ZEA-2-SIP04-V05), a function generator (Keysight
33511B), triaxial cables and a computer. A detailed description of the SIP measurement system and the specialized
sample holder are provided in Ehosioke et al. (2023).
The SIP measurement is performed by injecting alternating current at different frequencies (1 Hz – 45 kHz) into a
sample and measuring the amplitude and phase lag of the resulting voltage, which leads to a frequency dependent
electrical impedance expressed as:
$$Z_\omega^* = Z_\omega' + jZ_\omega'' \qquad\qquad (1)$$
where $Z_\omega^*$ is the complex impedance,  $\omega$ is the angular frequency, $Z'$ and $Z''$ are the real and imaginary parts of
the complex impedance, and $j$ is the imaginary unit. The complex impedance can be converted into the complex



electrical conductivity or electrical resistivity by accounting for the dimension of the sample using a geometric
factor (K = $\frac{\pi d^2}{4l}$ where $d$ is the root diameter and $l$ is the root length):
$\rho_\omega^* = KZ_\omega^* = |\rho|e^{j\varphi}$                  (2)
where $\varphi$ is the phase shift and $|\rho|$ is the resistivity magnitude. The relationship between complex conductivity $\sigma_\omega^*$
and complex resistivity $\rho_\omega^*$ is:
$\sigma_\omega^* = \frac{1}{\rho_\omega^*}$                  (3)

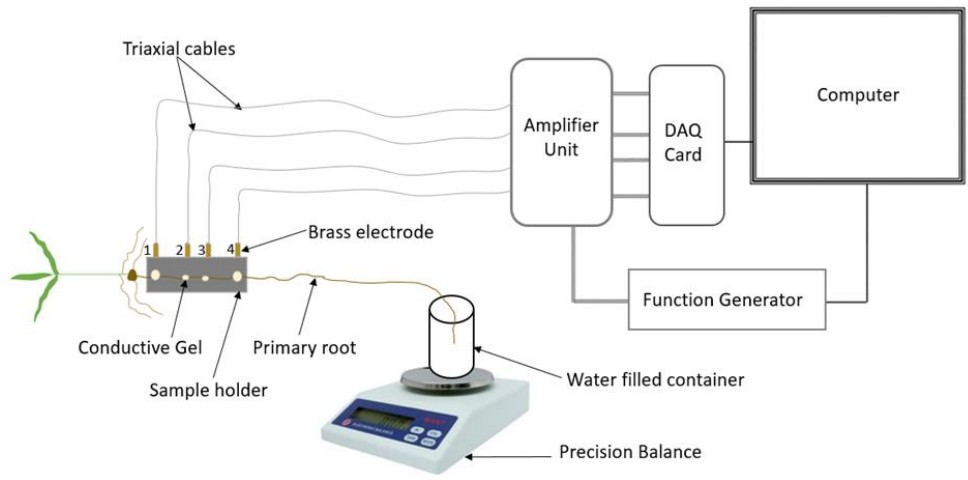


**Figure 2.** Measurement set-up for investigating the electrical response of roots during water uptake.

**2.3. Measurement protocol**
First, preliminary SIP measurements were performed on roots of Maize and Brachypodium plants in air to
investigate the effect of root drying on the SIP response. For this, one plant of each species was sampled. The root
was mounted in the sample holder and SIP measurements were taken at 5 minutes intervals for a total duration of
20 minutes with the root in the same position (see Fig. 2).
For water and salt uptake, the root was mounted on the sample holder and initial SIP measurement performed that
forms the baseline, before the root apex was tipped into a 50 ml demineralized water (e.g. Rewald et al. 2011; Li
et al. 2016) or saline water of known conductivity in a 60 ml sampling container (Fig. 2), and the initial weight of
the water, the container and the root tip was recorded. The weight was also recorded every 5 minutes for a total
duration of 20 minutes. Temperature and humidity were recorded at the end of the experiment. In the case of water



uptake, SIP measurements were acquired on one plant for each species using the same measurement strategy to
serve as a reference to help interpret the electrical response of roots to the uptake of salt solutions.
The SIP response of roots in different salt solutions was investigated in two experiments. In a first experiment, we
exposed one plant of each species to two different salt solutions i.e salt-L and salt-H (see Table 1). The SIP
measurements were performed at a 5 minutes interval over a 20 minutes duration while the root apex was tipped
in salt solution. In the second experiment, the effect of varying salt concentrations on the SIP response of the roots
was investigated. To achieve this, the measurement procedure described above was repeated with 7 different salt
solutions for Brachypodium (1000 – 10000 ppm) and another 7 different salt solutions for Maize (16800 – 60000
ppm) (see Table 1). Thus, a total of 14 plants was used in this experiment. To estimate evaporation loss during SIP
measurements, a 50 ml demineralized water was left open on the balance and the mass was measured every 5
minutes over a 20 minutes duration, this procedure was repeated for the salt solutions to estimate the loss of water
from the container due to evaporation. The evaporation loss was found to be 40 mg in 20 minutes for both
demineralized and saline water. The temperature and humidity at the time of measurement was also recorded (see
Appendix: Table B1). The net amount of solution absorbed by the root during each measurement corresponds to
the weight difference corrected for the estimated loss by evaporation.
**3. Results and Discussion**
**3.1. SIP monitoring of root dessication**
The resistivity magnitude and phase of exposed Brachypodium and Maize roots are shown in Fig. 3. We can
observe that the resistivity values of root segments of both species increased when the roots were exposed in the
air. Water content plays a key role in maintaining the structural properties and physiological processes of the cell
membrane (Crowe and Crowe 1982). Loss of water from roots may lead to a loss of turgor pressure (plasmolysis),
which can result in a decrease in cell volume depending on cell wall hardness (Verslues et al. 2006; Robbins and
Dinneny 2015), a decrease in cell membrane surface area, and cell membrane injury in severe cases (Lew 1996;
Ando et al. 2014). Wu et al. (2008) reported an increase in total impedance during dehydration of eggplant pulp.
Islam et al. (2019) also observed an increase in total impedance of onions during drying over a period of 21 days.
They concluded that movement of ions due to dehydration is responsible for the increased impedance. The increase
in resistivity observed in our studies for Maize and Brachypodium roots is due to loss of water from the root cells
(dehydration) due to evaporation. The increase in resistivity is higher for Brachypodium (78 Ωm increase in 20
minutes after the baseline measurement of 68 Ωm) than for Maize (7 Ωm increase in 20 minutes after a baseline
measurement of 16 Ωm) both in absolute and relative values. This suggests that Brachypodium root lost water



faster than Maize in our experiment. We had expected that Maize would lose more water because of the larger
surface area, but the result suggests that something other than surface area influenced the root dehydration, which
could be the degree of saturation. Since Maize roots were observed to be   more saturated than Brachypodium
roots in this study, it should take longer for Maize roots to lose sufficient water and become plasmolyzed compared
to Brachypodium roots. Shrinkage of Brachypodium root was clearly visible at the end of the measurement,
whereas Maize appeared dry on the surface but showed no significant shrinkage. The more noisy data observed
for Brachypodium is attributed to the high contact impedance of the root induced by shrinkage of Brachypodium
root during drying. Polarization (phase peak) of Brachypodium showed a decrease and a shift towards lower
frequencies while that of Maize first showed an increase followed by a stabilization. In a plasmolyzed cell, cell
membranes shrink (see Fig. 1), which is expected to result in a decrease of the phase response. It seems that
Brachypodium roots might have become plasmolyzed due to water loss (Lew 1996; Ando et al. 2014; Robbins and
Dinneney 2014), while Maize roots were not plasmolyzed but rather experienced osmotic adjustment by
redistribution of water to maintain equilibrium. This might explain why the phase response of Maize did not
decrease. It is important to note that during the dessication test, the leaves of both plants did not show any sign of
wilting (see Appendix C, Figure C1a and C2a).

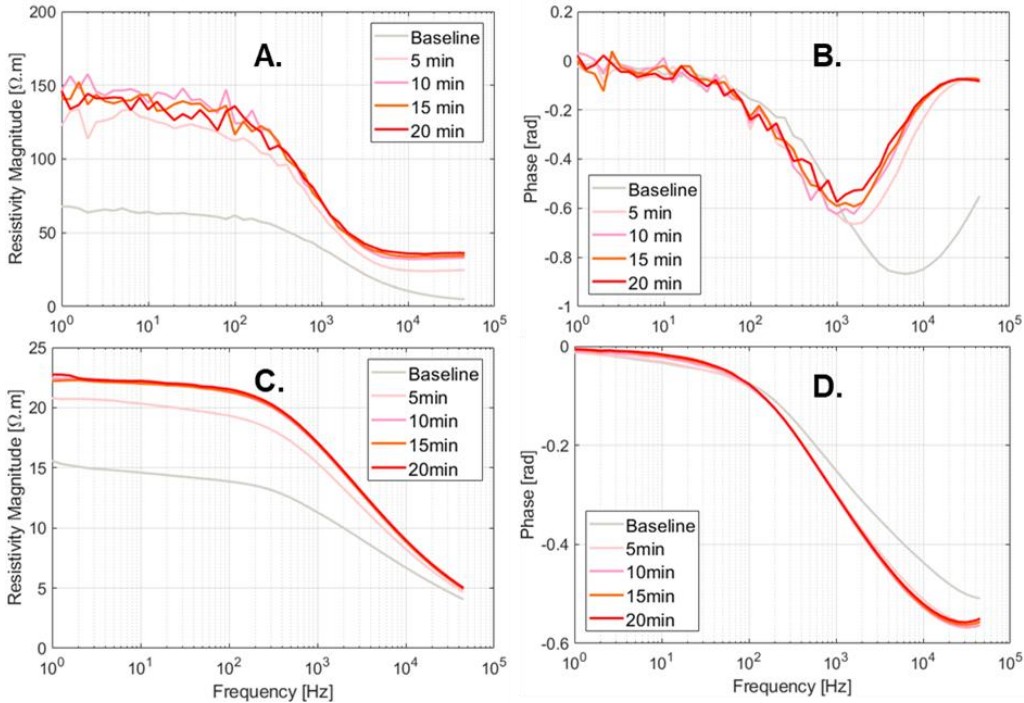


**Figure 3.** Resistivity and phase response of Brachypodium (**a-b**) and Maize (**c-d**) primary roots to drying.



**3.2. SIP monitoring of roots with their tips in demineralized water**
The change in mass of demineralized (DM) water during SIP measurements on Brachypodium and Maize roots is
shown in Table 2 and 3 respectively. The net mass of water uptake by the roots after correcting for evaporation
loss were 40 mg and 70 mg for Brachypodium and Maize root, respectively. The Maize absorbed more water
compared to Brachypodium since its leaf surface area is larger and thus has a larger transpiration pull.
**Table 2** Uptake of demineralized water and saline water by Brachypodium root in 20 minutes

| Time (min) | Mass (mg) | | |
|---|---|---|---|
| | Demin water | Salt-L | Salt-H |
| 0 | - | - | - |
| 5 | 20 | 20 | 20 |
| 10 | 20 | 20 | 20 |
| 15 | 20 | 20 | 20 |
| 20 | 20 | 30 | 20 |


**Table 3** Uptake of demineralized water and saline water by Maize root in 20 minutes

| Time (min) | Mass (mg) | | |
|---|---|---|---|
| | Demin water | Salt-L | Salt-H |
| 0 | - | - | - |
| 5 | 20 | 40 | 30 |
| 10 | 30 | 20 | 30 |
| 15 | 30 | 20 | 30 |
| 20 | 30 | 30 | 20 |


For both species, the resistivity magnitude shows an increase with a greater effect at low frequencies (< 1 kHz)
and almost no effect at high frequencies (> 10 kHz) for Maize (Fig. 4). According to the conduction mechanisms
illustrated in Fig. 1, this suggests that extracellular fluid is diluted by DM water, which results in the observed
higher resistivity. Polarization (phase peak) of Brachypodium showed no clear trend while that of Maize remained
mostly constant after an initial increase for a broad range of frequencies (10 to 10 000 Hz), which is consistent





with its resistivity magnitude. Uptake of DM water may lead to dilution of cellular solutes (Schopfer 2006), which
can decrease the water potential gradient across the cell membrane that drives water movement (Robbins and
Dinneny 2015). This adjustment will be reflected in the transmembrane potential leading to the polarization effect,
and the phase peak could reflect the water redistribution and equilibrium reached as the cell regains full turgor.
The phase response of Brachypodium root might be linked to the adjustment of the transmembrane potential while
the steady increase in phase response of Maize suggests that its transmembrane potential might be in equilibrium.

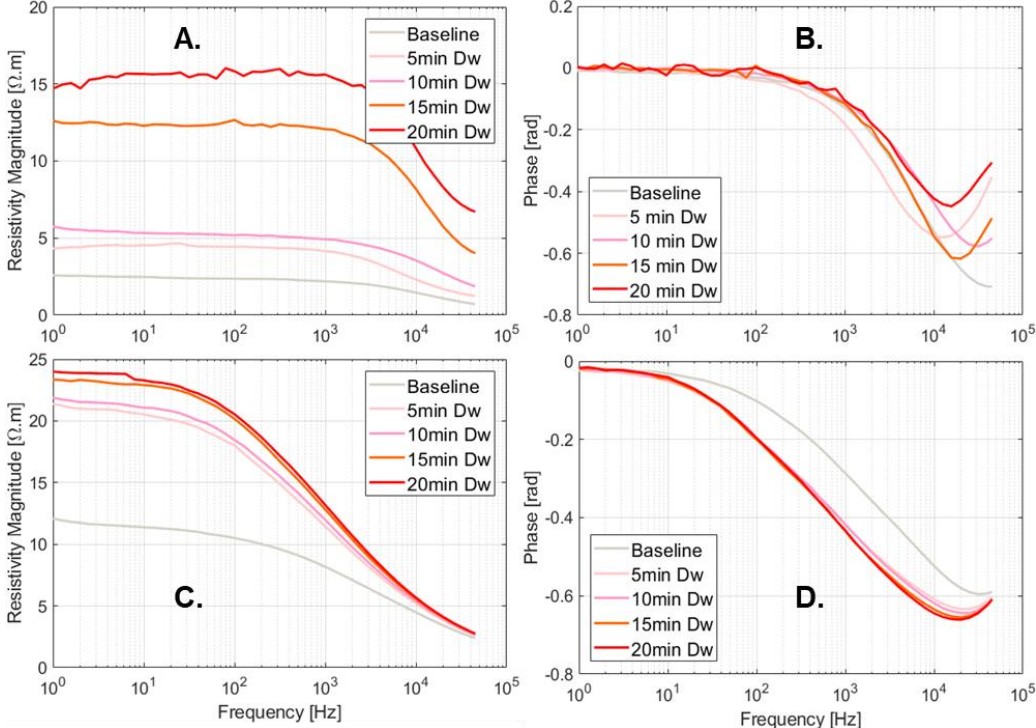


**Figure 4.** Resistivity magnitude and phase spectra of Brachypodium (**a-b**) and Maize (**c-d**) primary roots during absorption of
demineralized water. The variable temporal development of the resistivity magnitude might be due to high contact impedance
of the Brachypodium root.

**3.3. SIP monitoring of roots with their tips in saline water**
The net mass of saline water (salt-L/salt-H) absorbed by the roots was similar with 40/50 and 70/70 mg for
Brachypodium and Maize roots, respectively (Table 2 and 3). For the low salt concentration (Salt-L), the SIP
response of Maize (Fig. 5) showed a similar response as in the case of DM water with an increasing resistivity
magnitude and phase. In contrast, the Brachypodium root segments showed a continuous decrease of resistivity



magnitude and phase. This opposite behavior may be explained in terms of salt stress tolerance. Maize is known
to be moderately sensitive to salt stress (Farooq et al. 2015). Maize roots are able to take up water while excluding
salts, making it more robust to salinity stress (Neubert et al. 2005; Farooq et al. 2015; Munns et al. 2020). This
may explain why the SIP response of maize at this salt concentration level is similar to the response with DM
water. Apparently, the concentration of the salt-L solution was already too high for Brachypodium to exclude or
compartment salt in the vacuole (e.g. Lv et al. 2014) and the excess accumulation of ions in the root cell resulted
in the observed decrease in resistivity and polarization (phase peak). Additionally, after 20 minutes of measurement
with Brachypodium root tip in salt-L, the Brachypodium leaves showed visible signs of wilting (Appendix C:
Figure C2b) which is a key sign of salt toxicity in plants (e.g. Ji et al. 2022; Plant Ditech 2023). Similar signs of
wilting of leaves was observed in Maize leaves after 20 minutes of measurement with the root tip in saline water
of 40000 ppm (684 mM) (see Appendix C: Figure C1b). Drought is also known to cause wilting of leaves (e.g.
UCANR, 2021; Ji et al. 2022; PlantDitech 2023; Bayer 2024), however, the absence of wilting when the root tip
is not in saline solution for the same duration confirms that the wilting observed in this study is a clear indication
that the plants experienced salt toxicity.

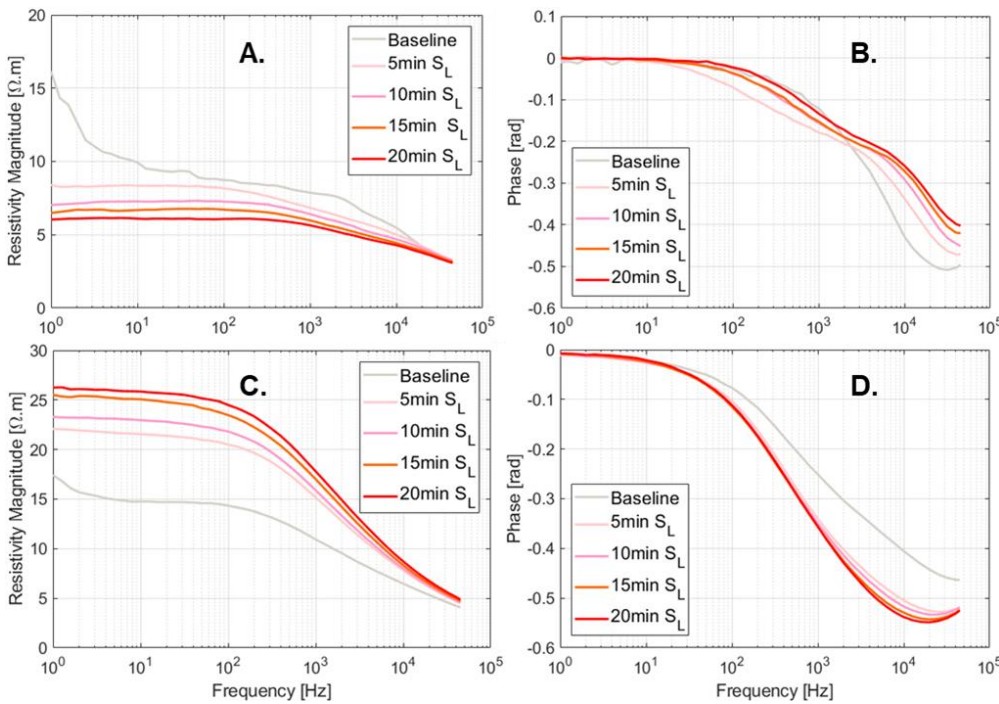


**Figure 5.** Changes in resistivity magnitude and phase spectra of Brachypodium (**a-b**) and Maize (**c-d**) primary roots during
absorption of saline water (salt-L).





During high salt concentration (salt-H) uptake (Fig.6), it is interesting to see that both Maize and Brachypodium
roots now have similar responses, showing a consistent decrease in both resistivity magnitude and phase. The
consistent decrease in resistivity magnitude and phase for both species suggests excessive accumulation of ions in
the cytoplasm and apoplast, which makes the roots more conductive (Debez et al. 2004; Ellouzi et al. 2011). At
this high salt concentration (Salt-H), the plant cells apparently cannot exclude all the sodium and chloride ions or
compartment them in the vacuole. This is probably the beginning of toxicity effects, although it will take time for
the damage to be visible. This early detection of ion toxicity is a key advantage of SIP for root salinity studies
(Ben Hamed et al. 2016).

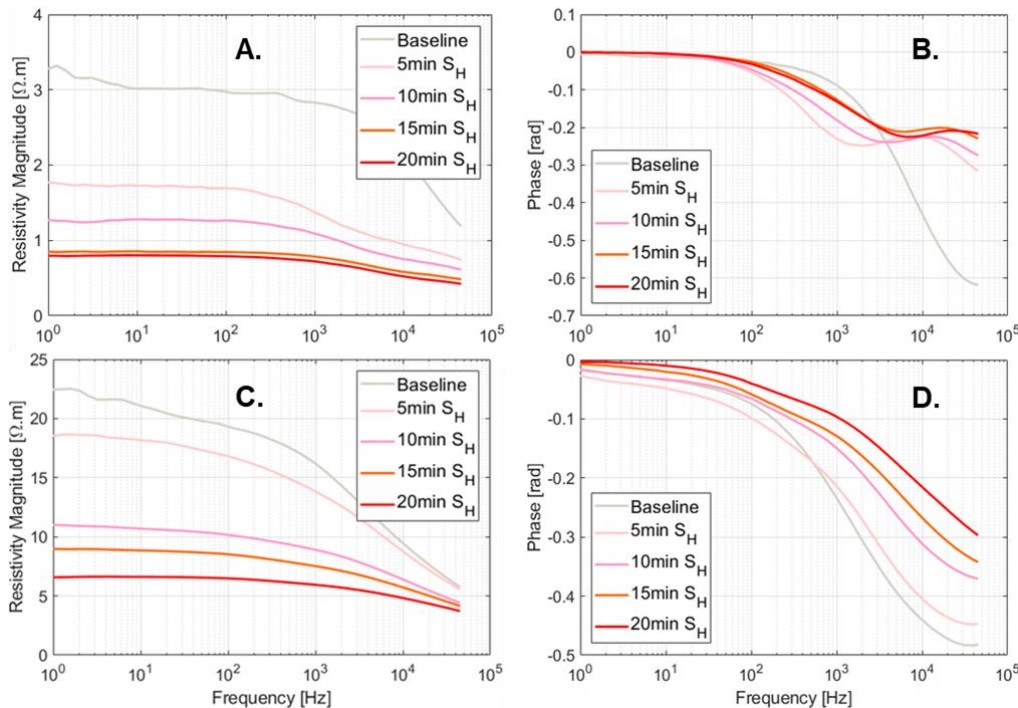


**Figure 6.** Changes in resistivity and phase spectra of Brachypodium (**a-b**) and Maize (**c-d**) primary roots during absorption of
saline water (salt-H).

**3.4. SIP monitoring of roots taking up water of gradually increasing salinity**
The SIP response of Maize and Brachypodium roots to increasing salinity is presented in Fig. 7. Note that the
range of salinity used for both species is different due to their different tolerance to salt stress. In general, a similar
resistivity response was observed for both species (Fig. 7a and 7c), showing either an increase or a decrease of
resistivity depending on the solute concentration, but with a different threshold due to their different salt stress





tolerance. For Maize roots, the phase response is similar to the resistivity response showing either an increase or
decrease with concentration over time (Fig. 7b) for a concentration threshold between 34000 and 35000 ppm. For
Brachypodium roots, a decrease of phase is observed at all concentrations after 10 minutes (Fig. 7d). Only at low
concentration (below 4000 ppm), an initial increase in phase was observed in the first 10 minutes of the experiment.

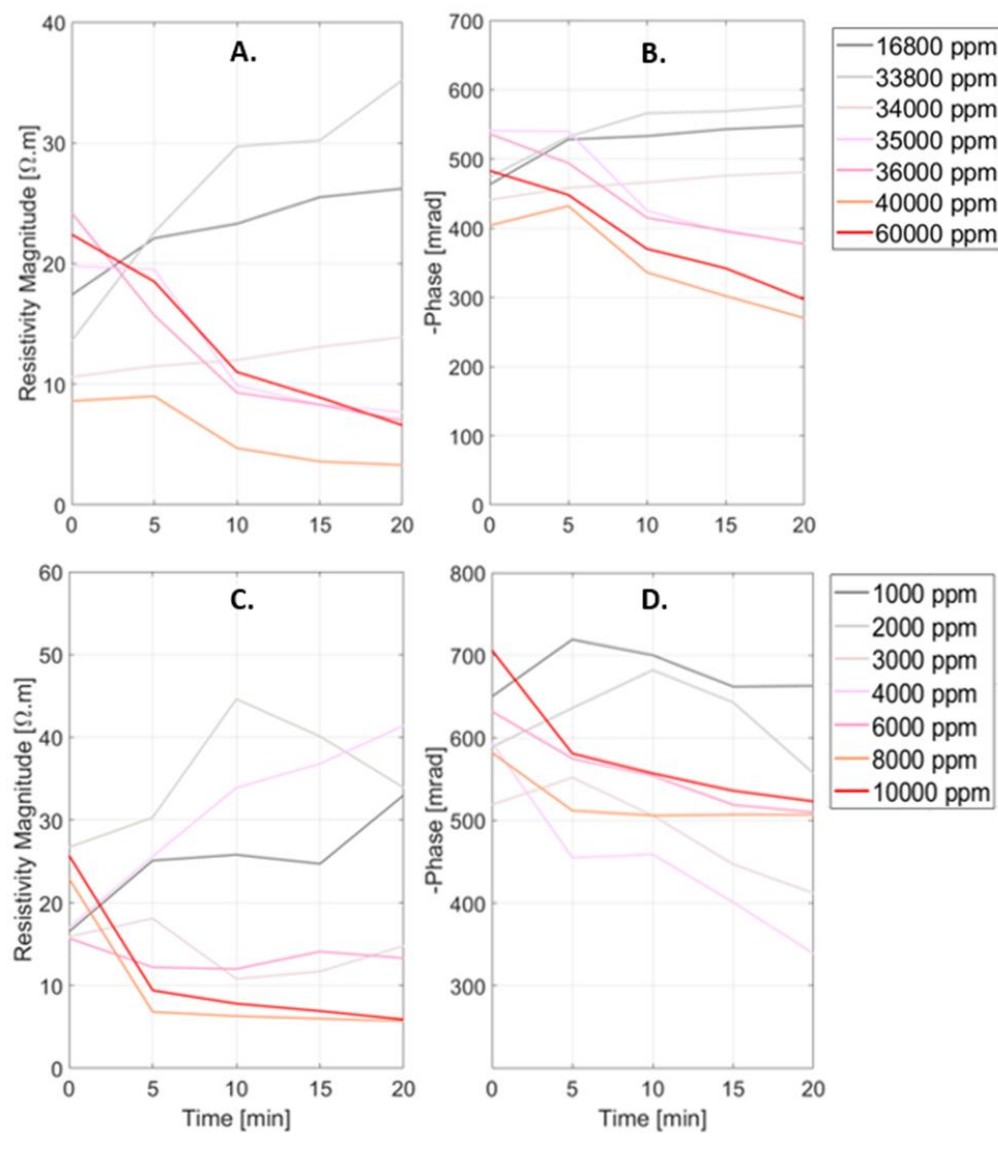


**Figure 7.** Changes in resistivity magnitude and phase peak of primary roots of Maize (**a-b**) and Brachypodium (**c-d**) with
concentration over time.



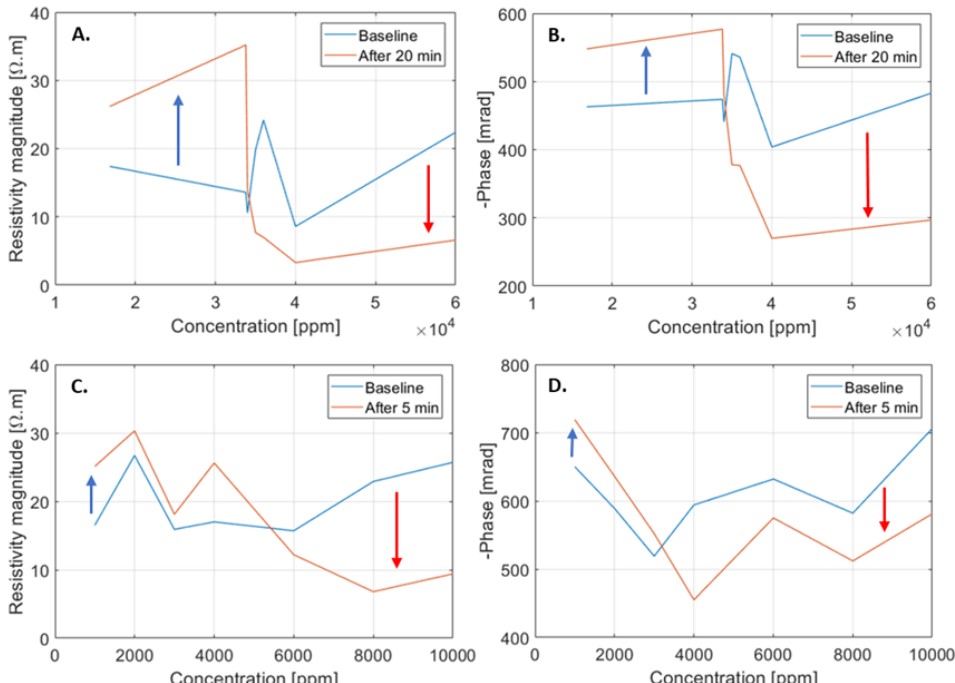


**Figure 8.** Reversal of resistivity magnitude and phase peak of Maize (**a-b**) and Brachypodium (**c-d**) primary roots as
concentration increases.

The adaptive mechanisms to salt stress may explain why the resistivity and phase response of the roots increased
at low salt concentrations and decreased at high salt concentration (Fig. 8). With increasing salt concentration,
excessive sodium accumulation in the cells occurs when the salt resistance threshold of the plant species is
exceeded (Cramer 1988; Davenport et al. 2005; Zhao et al. 2010; Farooq et al. 2015; Isayenkov and Maathuis
2019). Excess ions in the cell will increase the conductivity of the cellular fluid leading to decreased resistivity
and phase (e.g. Fig. 7 and 8). The disparity between the phase response of Maize root and Brachypodium root with
increasing salinity may be related to the salt resistance mechanisms of the species. These results seem to confirm
that Maize is more tolerant to salinity than Brachypodium, showing increasing resistivity and phase response up
to 34000 ppm before decreasing (Fig. 8a and 8b) while the Brachypodium show increasing resistivity only up to
5800 ppm before decreasing (Fig. 8c). The reversal of phase response in Brachypodium occurs at 3000 ppm but it
is only visible in the first 5 minutes (Fig. 8d). The threshold at which the reversal occurs in Maize falls within the
range of very highly saline water, while that of Brachypodium lies in the range of moderately saline water (see
Table 2).



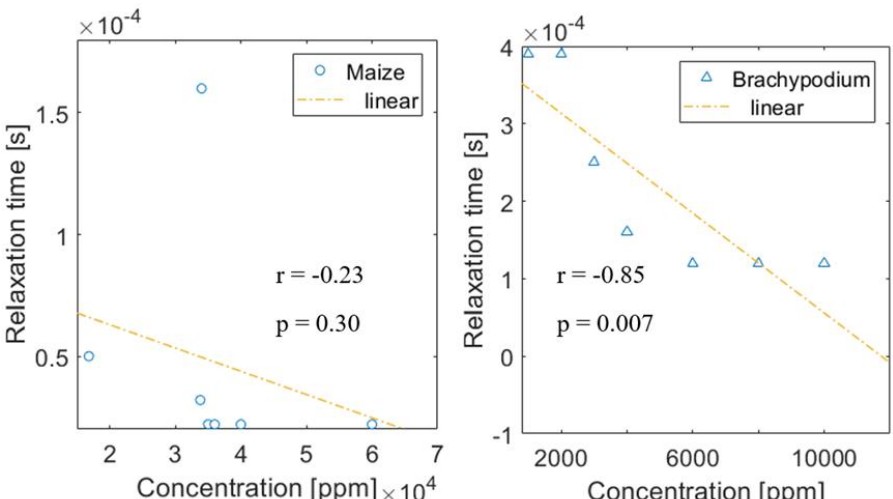


**Figure 9.** Correlation of relaxation time with NaCl concentration for Maize and Brachypodium primary roots. The relaxation
time $\tau_{max}$ is expressed as the inverse of $\omega_{max}$, where $\omega_{max}$ is the angular frequency at which the maximum phase shift occurs.


In Figure 9, we present a trend analysis of the relaxation time ($\tau_{max}$) and salt concentration during the reversal of
electrical response observed in Brachypodium (5 minutes) and Maize (20 minutes) as reported in Figure 8. Bücker
and Hördt (2013) reported that relaxation times are only weakly dependent on salinity in the case of pore radii, but
in this study we found a significant correlation between relaxation time and NaCl concentration in Brachypodium,
with Pearson's r = -0.85 and p value = 0.007. This further suggests that both species respond differently to salt
stress based on their salinity tolerance.
Salinity tolerance varies widely across plant species and even across genotypes within a species (Grieve et al.
2012). Thus, salinity tolerance of any plant is therefore indicated by the point or range in the continuum of salt
stress where visible or quantitative adverse effects are observed (Lauchli and Grattan 2012). In this study, the
concentration at which the reversal occurs for each species could be an indication of the salt resistance threshold
of the species (Grieve et al. 2012). This implies that salt tolerant species can withstand higher degrees of salinity
over a longer period of time.



**4. Conclusions**

We showed that SIP is able to detect the uptake of water and saline water in both Maize and Brachypodium roots, and that the conduction and polarization of Maize and Brachypodium roots were influenced by the degree of salinity. Plants respond to salt stress by excluding the ions from entering the cells (ion exclusion) and by removing the sodium and chloride ions from the cytoplasm and accumulating them in the vacuole (ion compartmentation). At relatively low salt concentration, the plants activate these salt resistance mechanisms leading to osmotic adjustment which helps the cells to maintain ionic balance, turgor and volume so that the plant can function optimally, which we observe as increasing resistivity and phase in the SIP signal. At very high salt concentration, there are more ions in the solution than the plant can exclude or compartment, which leads to excess sodium and chloride ions in the cytoplasm and apoplast (ion toxicity) which we observed as decreasing resistivity and polarization. The duration of salt stress and the salt concentration determine how long it takes for ion accumulation in plants to reach toxic levels. At very low concentrations, it might take days to weeks, but at very high concentrations it takes minutes only.

More studies should focus on testing the use of SIP method for early detection of salt stress in field grown crops. Future studies should be carried out with halophytes with a clear salt tolerance threshold,it would be interesting to know if the reversal of electrical properties at certain salt concentrations will match clearly with the salt tolerance threshold of the plants. In this study, we focused on single root segments (primary roots) in the laboratory. For field measurement, we sugest the use of an electrode set up that can be used to perform SIP measurements directly on the crop stem, which will solve the problem of current leakage through the soil-root interface in the case of stem-soil electrodes set up where the soil is more conductive than the roots (e.g. in a salty soil). Since the measurement at the root collar in this study detected uptake of saline water by the root tip, we expect that measurement at the root stem will also detect uptake of salt by the roots under field conditions.

**Appendices**

**Appendix A: Saline water classification**

**Table A1** Classification of saline water modified after Rhoades et al. (1992).

| Water classification | Salt concentration (ppm) | Electrical conductivity (mS/cm) |
|---|---|---|
|  |  |  |



| Non-saline | < 500 | 0.7 |
|---|---|---|
| Slightly saline | 500 - 1500 | 0.7 - 2 |
| Moderately saline | 1500 - 7000 | 2 - 10 |
| Highly saline | 7000 - 15000 | 10 - 25 |
| Very highly saline | 15000 - 35000 | 25 - 45 |
| Brine | > 35000 | > 45 |


**Appendix B: raw data from the experiments**


**Table B1** Evaporation estimation for demineralized water and salt solutions (salt-L and salt-H).

| Time(min) | Mass (g) | | | Temperature (°C) | | | Humidity (%) | | |
|---|---|---|---|---|---|---|---|---|---|
| | *D.water* | *Salt-L* | *Salt-H* | *D.water* | *Salt-L* | *Salt-H* | *D.water* | *Salt-L* | *Salt-H* |
| 0 | 54.08 | 55.24 | 57.27 | 26.7 | 26.5 | 26.2 | 36 | 32 | 30 |
| 5 | 54.07 | 55.23 | 57.27 | 26.5 | 26.5 | 26.6 | 36 | 32 | 31 |
| 10 | 54.06 | 55.22 | 57.25 | 26.9 | 26.5 | 27.0 | 36 | 32 | 30 |
| 15 | 54.05 | 55.21 | 57.24 | 27.1 | 26.6 | 27.4 | 36 | 32 | 30 |
| 20 | 54.04 | 55.20 | 57.23 | 27.3 | 26.6 | 28.2 | 36 | 32 | 28 |


**Table B2** Demineralized water uptake by Maize and Brachypodium in 20 minutes

| Time(min) | Mass (g) | | Temperature (°C) | |
|---|---|---|---|---|
| | *Maize* | *Brachypodium* | *Maize* | *Brachypodium* |
| 0 | 54.82 | 54.98 | 28.1 | 27.7 |
| 5 | 54.80 | 54.96 | 28.1 | 27.8 |
| 10 | 54.77 | 54.94 | 28.2 | 27.9 |





| | | | | |
|---|---|---|---|---|
| 15 | 54.74 | 54.92 | 28.2 | 27.9 |
| 20 | 54.71 | 54.90 | 28.3 | 28.0 |


**Table B3** Saline water uptake by Maize and Brachypodium roots in 20 minutes

| Time (min) | Salt-L | | | | Salt-H | | | |
|---|---|---|---|---|---|---|---|---|
| | Maize | | Brachypodium | | Maize | | Brachypodium | |
| | Mass (g) | Temp (°C) | Mass (g) | Temp (°C) | Mass (g) | Temp (°C) | Mass (g) | Temp (°C) |
| 0 | 55.54 | 26.1 | 55.71 | 26.2 | 57.66 | 26.4 | 57.79 | 26.8 |
| 5 | 55.50 | 26.6 | 55.69 | 26.6 | 57.63 | 26.4 | 57.77 | 26.8 |
| 10 | 55.48 | 26.7 | 55.67 | 26.9 | 57.60 | 26.6 | 57.75 | 26.8 |
| 15 | 55.46 | 26.8 | 55.65 | 27.0 | 57.57 | 26.9 | 57.73 | 26.9 |
| 20 | 55.43 | 26.7 | 55.62 | 26.9 | 57.55 | 27.1 | 57.71 | 26.9 |


**Appendix C: visual inspection of plants during the experiments**

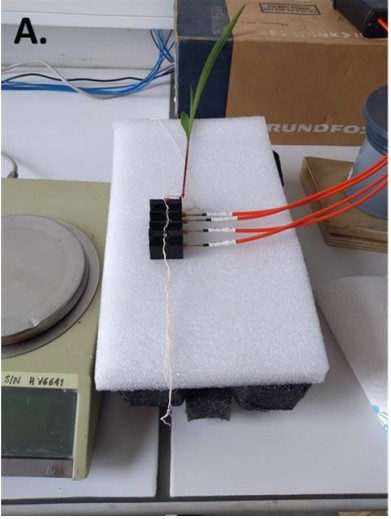
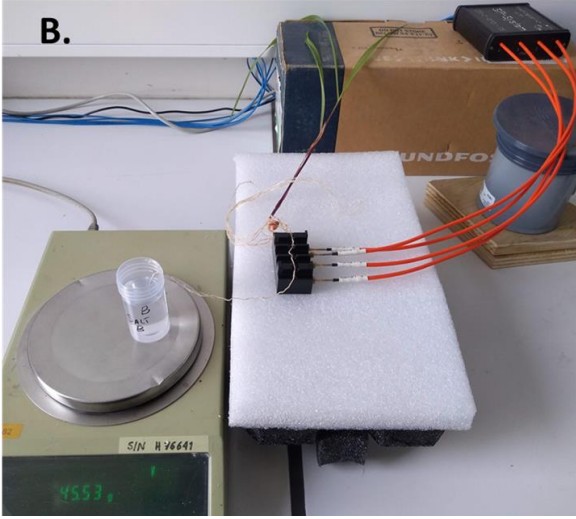

**Figure C1.** (a) Maize roots exposed during dessication test over 20 minute duration, the leaves showed no sign of wilting. (b)
Maize roots exposed with the primary root tip in saline water of 40000 ppm (684 mM) concentration, the leaves showed
visible signs of wilting after 20 minutes of measurement.



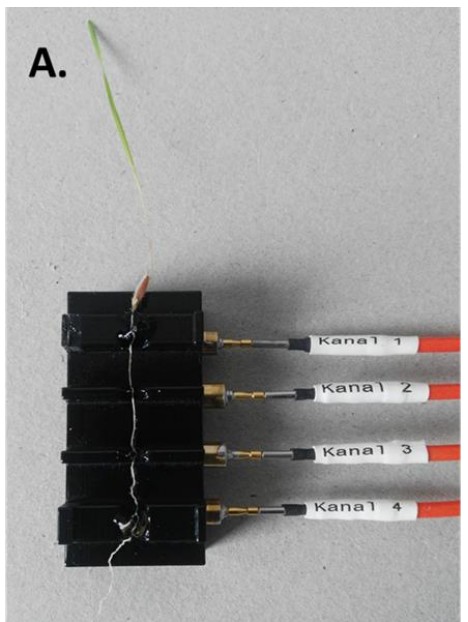 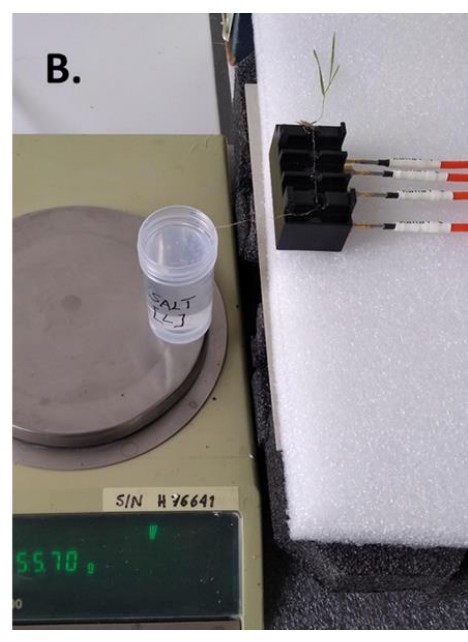


**Figure C2.** (a) Brachypodium root exposed during dessication tests over 20 minute duration, the leaves showed no sign of
wilting. (b) Brachypodium roots exposed with the primary root tip in salt-L solution of 16800 ppm (287 mM) concentration,
the leaves showed visible signs of wilting after 20 minutes of measurement.


**Author Contributions**
Conceptualization: SE, FN, SG & MJ
Methodology: SE, FN, JAH, & EZ
Data curation, analysis and visualization: SE, JAH, FN, & EZ
Original draft: SE
Review and editing: All authors
Funding acquisition: SG, FN & MJ
Supervision: SG, FN, MJ & JAH

**Conflict of Interest**
The authors declare no conflict of interest

**Data Availability Statement**
Data associated with this study will be made available on request.



**Acknowledgements**
This research was funded by the Belgian National Fund for Scientific Research FNRS (F.R.S-FNRS).

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
