# Peer review of "Technical note: Investigating saline water uptake by roots using"

_EGUsphere, 2024_

## Author Comment (AC1)

**Final Author Comments (ACs)**

**RC1: 'Comment on egusphere-2024-2628', Imre Cseresnyes, 25 Oct 2024**

**General comments:** High-quality paper with actual choice of topic and important novelties. Congratulations for the Authors!

**Reply:** Thanks for your careful and detailed review, and for providing interesting comments and suggestions that would greatly improve the manuscript.

**Specific comments:**

1) L36: Consider to add "electrical impedance" and/or "phase angle" to the keywords.

**Reply:** "electrical impedance" and "phase angle" will be added to the keywords in the revised version.

2) L42–44: Besides low water potential and ion toxicity, salinity provokes an oxidative stress as well by excessive ROS generation. Consider to mention it as a third effect.

**Reply:** Thanks for pointing this out, it would be included in the revised version

3) L55: Among plant physiological processes, diurnal cycles in root uptake activity has recently been monitored by impedance measurement, see: https://doi.org/10.1186/s13007-023-01133-8

**Reply:** The authors have read this paper and agree that it is worth mentioning here, it will be included in the revised version

4) L59: Basically, there is a composite water pathway inside the root cylinder, including three routes temporally variable: 1) apoplastic pathway (cell walls and extracellulars), 2) symplastic pathway (plasmodesmata), 3) transcellular pathway through (aquaporin channels). The last two are often named "cell-to-cell pathway".

**Reply:** As you rightly noted, "cell to cell pathway" or "intercellular pathway" is often used to refer to the last two above. This was the intention of the authors. L59 would be revised to read "intercellular (plasmodesmata and aquaporins)…"

5) L68–70: It is worth mentioning that living tissues are equivalent to parallel RC circuits, which has a characteristic phase angle depending on AC frequency. It could be important later.

**Reply:** The authors agree and would add this in the revised version

6) L71: There is another work to evaluate salinity effect on impedance phase angle at a single frequency: https://doi.org/10.1016/j.biosystemseng.2018.03.004

**Reply:** Having read the suggested article, the authors agree that it is very relevant to the topic and would cite it here too.

7) L74 (caption of Fig. 1): "Low" and "high" frequency is rather relative. I think that a specification of frequency ranges (according to alpha and beta dispersion regions) would give a help for the readers.

**Reply:** Thanks for your suggestion. While it appears that specifying frequency ranges would be helpful here, there is no "specific frequency threshold" that controls the current pathway in all

plants. The frequency at which the current pathway changes is expected to vary between one plant and another depending on the properties of their cell membranes.

We agree that the alpha, beta and gamma dispersion ranges are used to differentiate the three polarization mechanisms in biomaterials such as plant roots, but the dispersion ranges overlap, which explains why different polarization processes are often found within a dispersion range that could not be explained using one time constant model.

8) L87–104: Consider to shorten the description of the previous work by Ben Hamed et al. (2016), focusing the main finding only. I think this long description is not necessary.

**Reply:** The authors will shorten this paragraph where necessary in the revised version

9) L113–114: "More studies are still needed to better understand how roots respond to salt stress." I fully agree with it, as root cells are the first target of soil salinity. I may be emphasized here.

**Reply:** Yes, we will emphasize this as suggested.

10) L123–125: As maize tolerance to salinity depends on genotype (as you write), specify the cultivar of maize applied in the experiment, and add some information (if available) of its salinity tolerance level.

**Reply:** Thank you for this suggestion, we agree that it would be nice to specify the cultivar and it's salinity threshold (if available). We tried to retrieve this information but unfortunately we were not successful, mainly because this study was conducted back in 2019 and we lost track of the exact source of the maize seeds.

11) L146: Add terminal (input) voltage of the AC signal used for measurement.

**Reply:** The input voltage used is 5V, this will be added to the text here during revision

12) L147 and thereafter: In my opinion, it would be better to always use the conventional symbol φ (phí) for phase angle both in the text and in the figures. Likewise, symbol "R" is worth using for the magnitude of resistance.

**Reply:** Thanks for your suggestion. The authors think that the use of "phase" or the symbol "φ" is mainly a matter of style as both means exactly same.

Also, in the context of SIP, we measure complex impedance (see L149) and use the geometric factor to compute the complex resistivity magnitude which we showed, the use of the symbol "R" does not apply in this case.

13) L208–209: "Polarization (phase peak) of Brachypodium showed a decrease and a shift towards lower frequencies while that of Maize first showed an increase followed by a stabilization." The sentence is difficult to follow. Make clear that changes occurred over root exposition time, and consider to give frequencies at which the phase shift reach a peak.

**Reply:** We agree, this will be clarified during revision

14) L222: I think "larger canopy transpiration" could be written instead of "larger transpiration pull". The latter characterize the negative xylem pressure, which was not obviously higher in maize.

**Reply:** Thanks for the suggestion, L222 will be revised to read "larger canopy transpiration"

15) L230–231: "Polarization (phase peak) of Brachypodium showed no clear trend while that of Maize remained mostly constant after an initial increase for a broad range of frequencies" For clarity,

supplement the sentence that there was a temporal trend, according to the absorption time of DM water.

**Reply:** The authors agree with your suggestion and will revise the paragraph accordingly

16) L258–259: "Drought is also known to cause wilting of leaves (e.g. UCANR, 2021; Ji et al. 2022; PlantDitech 2023; Bayer 2024)…" This is evident, references are not necessary, and should be deleted from here.

**Reply:** We agree that references are not necessary here, and will delete them during the revision

17) L266–268: "The consistent decrease in resistivity magnitude and phase for both species suggests excessive accumulation of ions in the cytoplasm and apoplast, which makes the roots more conductive" Additionally, salinity can lead to membrane damage with increased permeability (https://doi.org/10.1016/j.biosystemseng.2018.03.004). I think this also contributed to the changes observed in the present study.

**Reply:** Membrane damage was mentioned earlier (L193), linked with root desiccation. It makes sense that salinity could lead to membrane damage due to increased permeability

18) L300–302: Add literature, if available, to show the salinity thresholds tolerated by some maize genotypes.

**Reply:** Some Maize genotypes tolerate up to 6 mS/cm (e.g. Islam et al. 2024). This will be included in the revision

19) Fig. 9: For maize, one data seems to be an outlier. Have you tested the correlation without it? Perhaps it would be improved.

**Reply:** Thanks for your observation, we will do that during the revision.

**Technical corrections:**

1) Begin a new paragraph from L55.

**Reply:** OK

2) Write *Brachypodium* in italics.

**Reply:** OK

3) Write "maize" with lowercase letter, not capital.

**Reply:** OK

4) Fig 2–7: Using more contrasting colors for the curves may improve the visibility of the results.

**Reply:** We will edit figures 2-7 with more contrasting colors

5) Consider to merge Table 2 and 3.

**Reply:** Table 2 and 3 will be merged in the revised version

6) Fig. 7–9: It is confusing that the ranking of the two species (maize a-b, Brachypodium c-d) is the opposite to those of the previous figures. It is worth changing them.

**Reply:** Thanks for the observation, we will change the order as noted.

---

## Author Comment (AC2)

**Final Author comments (ACs)**

**RC2**: 'Comment on egusphere-2024-2628', Anonymous Referee #2, 05 Dec 2024

This manuscript addresses the suitability of Spectral Induced Polarization (SIP) for detecting salt stress in plant roots, using Brachypodium and Maize as model species. The topic is innovative, exploring a less-studied method for assessing root responses to salinity stress. While the study presents promising results, there are significant methodological and interpretative limitations that need to be addressed to enhance the robustness of the conclusions.

**Reply:** Thanks for your comment, we will revise the manuscript to address the limitations where necessary.

**Main Comments**

- The preliminary SIP measurements were performed on single plants for each species. Given the variability in biological systems, triplicate measurements are necessary to provide statistically meaningful baseline data.

**Reply:** Thanks for your comment. The baseline in figure 3-6 & 8 refers to the initial measurement on each plant before the root was tipped in water or saline water (L166-170), this enabled us to observe the change in SIP spectra due to the uptake of water or saline water for a duration of 20 minutes. It is therefore not necessary to measure several plants to establish the baseline in this case.

In general, we agree that triplicate measurement is useful in biological system, but we performed several trial measurements with replicates of maize and Brachypodium plants before the actual experiments reported here. In all the replicates, we observed similar response for drying, water and saline water uptake, and we have attached figures here for some of the replicate measurements (**see attached additional figures 1-4**), thus we argue that the issue of variability and reproducibility were taken into account in the experiment. These additional figures could be added to the appendix to help clarify the question of replicates.

- **Fig. 3:** If I understand correctly, you suggest that the initial increase in resistivity (from the baseline to 5 minutes) is attributed to water loss through evaporation from the root. However, the changes in resistivity between 5 and 20 minutes are much smaller. Could you elaborate on the factors contributing to the decrease in the evaporation rate during this period?

**Reply:** Before measurement, the plant was removed from the soil and placed on the sample holder, at this time the root is moist on the surface. The water loss to evaporation in the first 5 minutes is due to evaporation of the water film on the root surface. What we observe from 5-20 minutes is a response from within the root.

In addition, I assume that the SIP measurement duration is on the same order of magnitude as the times reported here. Can you clarify?

**Reply:** Each SIP measurement takes about 1.5-2 minutes to complete. After the baseline measurement, subsequent measurements were started exactly at 5 minutes intervals.

The Mettler PM 2000 balance has a weighing capacity of 2,100 grams with a readability of 10 mg, a reproducibility of 5 mg, and a linearity of ±20 mg. Given these specifications, measuring changes as small as 20 mg approaches the balance's linearity limit, potentially compromising the accuracy of such measurements. Therefore, the accuracy of the 20 mg changes reported in Tables 2 and 3 is questionable when using this balance.

**Reply:** Thank you for your suggestion. We agree that 20 mg is close to the linearity limit of the balance. However, we conducted several trial measurements with this PM 2000 balance prior to the reported results (see attached additional figures 1-4), and we are confident that our results are valid and reproducible (please see the attached additional figures).

However, in subsequent experiments, we will use a more robust balance to avoid similar challenges.

According to your data, the water uptake by the plant appears to be constant over time. How do you explain the nonconstant change in resistivity over time in light of this observation?

**Reply:** Thanks for your observation. This is due to a combination of two factors:

1. The drying out of the exposed root surface as in the case of desiccation test

2. The uptake of water by the root tip

The baseline measurement was performed with root exposed, then the root tip is placed in water, and measurements are taken at 5 minutes interval. We expect that the surface of the root will dry out first and increase the resistivity, then as the root takes up more water the resistivity becomes more stabilized.

It is also challenging to understand how water uptake by the plant is equal or higher than evaporation. Typically, transpiration represents a fraction of the evaporation from a free surface. In this case, the free surface is the water in the tube, and the plant extracts water only from its root tip. Given the relatively small contact area between the root and the water, could you clarify

**Reply:** Thanks for your comment, this experiment was performed in a controlled environment where the temperature and humidity were kept relatively constant, with a focus to reducing evaporation in order to observe and quantify the uptake of water from a single root segment ( see L179-185). We argue that this outcome is actually expected.

Also, we measured evaporation from the tube without the root in it for same duration of the experiment (see Table B1 in appendix), this helped us to properly separate water uptake by the roots from the water loss due to evaporation.

I am concerned about interpreting physiological mechanisms based solely on SIP measurements without supporting data, such as direct measurements of root salt levels. Without corroborating evidence, these claims remain speculative and weaken the study's impact.

**Reply:** Thanks for your comment. While we did not directly measure salt levels in the root, we have evidence that wilting of leaves occurred during the salinity tests in 20 minutes but not during desiccation tests of the same 20 minutes duration (see appendix)

I recommend reporting such an experiment with repetitions, which is far more important than studying various salt levels at this stage.

**Reply:** For the saline water uptake, we sampled a total of 14 plants (L177-179), thus we argue that the issue of repetitions was accounted for here.

Also, we mentioned earlier that prior to the reported results, we repeated these measurements several times with different replicates of maize and Brachypodium roots and the response to desiccation, water and saline water uptake were all the same. We attached some of those results here and will include them in the appendix in the revised version.

**L203:** Could you clarify what you mean by "Maize roots were observed to be more saturated than Brachypodium"? How was the saturation level of the different plants assessed, and what criteria or methods were used to draw this comparison?

**Reply:** We used the wrong choice of words here, what we mean is that the sampled maize roots were observed to be succulent and white in color, while Brachypodium roots were dry and brownish in color.

We will update L203 in the revised version to read "the sampled maize roots were observed to be succulent and whitish in color, while Brachypodium roots were dry and brownish in color.."

**L212-214:** The statements made here are quite strong. Can you provide supporting evidence or references from the literature to substantiate these claims?

**Reply:** The statements on L212-214 will be toned down during the revision to read "Maize roots were probably not plasmolyzed but rather experienced osmotic adjustments".

The following literatures which described "osmotic adjustments" in Maize roots under stress will be cited here in the revised manuscript:

1. Sharp et al., 1990: https://doi.org/10.1104/pp.93.4.1337

2. Voetberg and Sharp, 1991: https://doi.org/10.1104/pp.96.4.1125

3. Ogawa and Yamauchi, 2006: https://doi.org/10.1626/pps.9.27

4. Hajlaoui et al., 2010: https://doi.org/10.1016/j.indcrop.2009.09.007

**Additional Figures**

**1. Water uptake**

**Maize root**

[Figure]

[Figure]

**Figure 1.** Resistivity and Phase response (a-b) of Maize during water uptake for 25 minutes. Measurement at 0 minute represents the baseline, measurement was repeated after 5 minutes (to observe drying effect) before putting the root tip in water at 10, 15, 20 and 25 minutes.

Brachypodium root

[Figure]

[Figure]

**Figure 2.** Resistivity and Phase response (a-b) of Brachypodium root during water uptake for 25 minutes. Measurement at 0 minute represents the baseline, measurement was repeated after 5 minutes (to observe drying effect) before putting the root tip in water at 10, 15, 20 and 25 minutes.

**Saline water uptake replicates**

Maize Root (salt-H)

[Figure]

[Figure]

**Figure 3.** Resistivity and Phase response (a-b) of Maize during saline water uptake (Salt-H) for 20 minutes. Measurement at 0 minute represents the baseline, measurement was repeated after 5 minutes to observe drying effect, before putting the root tip in water at 10, 15 and 20 minutes.

Brachypodium Root (Salt-L)

[Figure]

[Figure]

**Figure 4.** Resistivity and Phase response (a-b) of Brachypodium root during saline water uptake  (Salt-L) for 20 minutes. Measurement at 0 minute represents the baseline, measurement was repeated after 5 minutes, before putting the root tip in water at 10, 15 and 20 minutes.

---

## Author Response (AR1)

**Response to reviewer 1 comments**

**RC1:** **'Comment on egusphere-2024-2628',** **Imre Cseresnyes, 25 Oct 2024**

**General comments:** High-quality paper with actual choice of topic and important novelties. Congratulations for the Authors!

**Reply:** Thanks for your careful and detailed review, and for providing interesting comments and suggestions that would greatly improve the manuscript.

**Specific comments:**

1) L36: Consider to add "electrical impedance" and/or "phase angle" to the keywords.

**Reply:** "electrical impedance" and "phase angle" have been added to the keywords in the revised version. [See L36 of the marked-up manuscript]

2) L42–44: Besides low water potential and ion toxicity, salinity provokes an oxidative stress as well by excessive ROS generation. Consider to mention it as a third effect.

**Reply:** Thanks for pointing this out, it has been added [see L43-44]

3) L55: Among plant physiological processes, diurnal cycles in root uptake activity has recently been monitored by impedance measurement, see: https://doi.org/10.1186/s13007-023-01133-8

**Reply:** The authors have read this paper and agree that it is worth mentioning here, it is included in the revised version [L60]

4) L59: Basically, there is a composite water pathway inside the root cylinder, including three routes temporally variable: 1) apoplastic pathway (cell walls and extracellulars), 2) symplastic pathway (plasmodesmata), 3) transcellular pathway through (aquaporin channels). The last two are often named "cell-to-cell pathway".

**Reply:** As you rightly noted, "cell to cell pathway" or "intercellular pathway" is often used to refer to the last two above. This was the intention of the authors. We revised it to read "intercellular (plasmodesmata and aquaporins)…" see L62

5) L68–70: It is worth mentioning that living tissues are equivalent to parallel RC circuits, which has a characteristic phase angle depending on AC frequency. It could be important later.

**Reply:** The authors agree and have added this in the revised version [L70-72]

6) L71: There is another work to evaluate salinity effect on impedance phase angle at a single frequency: https://doi.org/10.1016/j.biosystemseng.2018.03.004

**Reply:** Having read the suggested article, the authors agree that it is very relevant to the topic and have cited it here too [see L75]

7) L74 (caption of Fig. 1): "Low" and "high" frequency is rather relative. I think that a specification of frequency ranges (according to alpha and beta dispersion regions) would give a help for the readers.

**Reply:** Thanks for your suggestion. While it appears that specifying frequency ranges would be helpful here, there is no "specific frequency threshold" that controls the current pathway in all

plants. The frequency at which the current pathway changes is expected to vary between one plant and another depending on the properties of their cell membranes.

We agree that the alpha, beta and gamma dispersion ranges are used to differentiate the three polarization mechanisms in biomaterials such as plant roots, but the dispersion ranges overlap, which explains why different polarization processes are often found within a dispersion range that could not be explained using one time constant model.

8) L87–104: Consider to shorten the description of the previous work by Ben Hamed et al. (2016), focusing the main finding only. I think this long description is not necessary.

Reply: The authors shortened this paragraph as suggested [ see marked-up manuscript L93-95 and L106-108]

9) L113–114: "More studies are still needed to better understand how roots respond to salt stress." I fully agree with it, as root cells are the first target of soil salinity. I may be emphasized here.

Reply: Yes, we emphasized this as suggested [L112-113]

10) L123–125: As maize tolerance to salinity depends on genotype (as you write), specify the cultivar of maize applied in the experiment, and add some information (if available) of its salinity tolerance level.

Reply: Thank you for this suggestion, we agree that it would be nice to specify the cultivar and it's salinity threshold (if available). We tried to retrieve this information but unfortunately we were not successful, mainly because this study was conducted back in 2019 and we lost track of the exact source of the maize seeds.

11) L146: Add terminal (input) voltage of the AC signal used for measurement.

Reply: The input voltage used is 5V, this has been added to the text here [L147]

12) L147 and thereafter: In my opinion, it would be better to always use the conventional symbol φ (phí) for phase angle both in the text and in the figures. Likewise, symbol "R" is worth using for the magnitude of resistance.

Reply: Thanks for your suggestion. The authors think that the use of "phase" or the symbol "φ" is mainly a matter of style as both means exactly same.

Also, in the context of SIP, we measure complex impedance (see L149) and use the geometric factor to compute the complex resistivity magnitude which we showed, the use of the symbol "R" does not apply in this case.

13) L208–209: "Polarization (phase peak) of Brachypodium showed a decrease and a shift towards lower frequencies while that of Maize first showed an increase followed by a stabilization." The sentence is difficult to follow. Make clear that changes occurred over root exposition time, and consider to give frequencies at which the phase shift reach a peak.

Reply: We agree, this has been clarified [L214-216]

14) L222: I think "larger canopy transpiration" could be written instead of "larger transpiration pull". The latter characterize the negative xylem pressure, which was not obviously higher in maize.

Reply: Thanks for the suggestion, this is now revised to read "larger canopy transpiration" [L231-232]

15) L230–231: "Polarization (phase peak) of Brachypodium showed no clear trend while that of Maize remained mostly constant after an initial increase for a broad range of frequencies" For clarity, supplement the sentence that there was a temporal trend, according to the absorption time of DM water.

**Reply:** The authors agree with your suggestion and have revised the paragraph accordingly [241-242]

16) L258–259: "Drought is also known to cause wilting of leaves (e.g. UCANR, 2021; Ji et al. 2022; PlantDitech 2023; Bayer 2024)…" This is evident, references are not necessary, and should be deleted from here.

**Reply:** We have deleted the references in question

17) L266–268: "The consistent decrease in resistivity magnitude and phase for both species suggests excessive accumulation of ions in the cytoplasm and apoplast, which makes the roots more conductive" Additionally, salinity can lead to membrane damage with increased permeability (https://doi.org/10.1016/j.biosystemseng.2018.03.004). I think this also contributed to the changes observed in the present study.

**Reply:** Membrane damage was mentioned earlier (L192), linked with root desiccation. It makes sense that salinity could lead to membrane damage due to increased permeability. We have now included this here [L286-287]

18) L300–302: Add literature, if available, to show the salinity thresholds tolerated by some maize genotypes.

**Reply:** We have now revised it as suggested [L320]

19) Fig. 9: For maize, one data seems to be an outlier. Have you tested the correlation without it? Perhaps it would be improved.

**Reply:** Thanks for your observation, we removed the outlier and the correlation improved.

**Technical corrections:**

1) Begin a new paragraph from L55.

**Reply:** Done

2) Write *Brachypodium* in italics.

**Reply:** Done

3) Write "maize" with lowercase letter, not capital.

**Reply:** Done

4) Fig 2–7: Using more contrasting colors for the curves may improve the visibility of the results.

**Reply:** We edited figures 2-7 with more contrasting colors

5) Consider to merge Table 2 and 3.

**Reply:** Table 2 and 3 has been merged

6) Fig. 7–9: It is confusing that the ranking of the two species (maize a-b, Brachypodium c-d) is the opposite to those of the previous figures. It is worth changing them.

**Reply:** Thanks for the observation, we changed the order as noted.

**Response to Reviewer 2**

**RC2**: 'Comment on egusphere-2024-2628', Anonymous Referee #2, 05 Dec 2024

This manuscript addresses the suitability of Spectral Induced Polarization (SIP) for detecting salt stress in plant roots, using Brachypodium and Maize as model species. The topic is innovative, exploring a less-studied method for assessing root responses to salinity stress. While the study presents promising results, there are significant methodological and interpretative limitations that need to be addressed to enhance the robustness of the conclusions.

**Reply:** Thanks for your comment, we revised the manuscript to address the limitations where necessary.

**Main Comments**

- The preliminary SIP measurements were performed on single plants for each species. Given the variability in biological systems, triplicate measurements are necessary to provide statistically meaningful baseline data.

**Reply:** Thanks for your comment. The baseline in figure 3-6 & 8 refers to the initial measurement on each plant before the root was tipped in water or saline water (L166-170), this enabled us to observe the change in SIP spectra due to the uptake of water or saline water for a duration of 20 minutes. It is therefore not necessary to measure several plants to establish the baseline in this case.

In general, we agree that triplicate measurement is useful in biological system, but we performed several trial measurements with replicates of maize and Brachypodium plants before the actual experiments reported here. In all the replicates, we observed similar response for drying, water and saline water uptake, and we have included additional figures in the appendix to show the replicate measurements (**see Appendix D: Figure D1, D2 and D3**), thus we argue that the issue of variability and reproducibility were addressed in the experiment.

- **Fig. 3:** If I understand correctly, you suggest that the initial increase in resistivity (from the baseline to 5 minutes) is attributed to water loss through evaporation from the root. However, the changes in resistivity between 5 and 20 minutes are much smaller. Could you elaborate on the factors contributing to the decrease in the evaporation rate during this period?

**Reply:** Before measurement, the plant was removed from the soil and placed on the sample holder, at this time the root is moist on the surface. The water loss to evaporation in the first 5 minutes is due to evaporation of the water film on the root surface. What we observe from 5-20 minutes is a response from within the root.

In addition, I assume that the SIP measurement duration is on the same order of magnitude as the times reported here. Can you clarify?

**Reply:** Each SIP measurement takes about 1.5-2 minutes to complete. After the baseline measurement, subsequent measurements were started exactly at 5 minutes intervals.

The Mettler PM 2000 balance has a weighing capacity of 2,100 grams with a readability of 10 mg, a reproducibility of 5 mg, and a linearity of ±20 mg. Given these specifications, measuring changes as small as 20 mg approaches the balance's linearity limit, potentially compromising the accuracy of such measurements. Therefore, the accuracy of the 20 mg changes reported in Tables 2 and 3 is questionable when using this balance.

**Reply:** Thank you for your suggestion. We agree that 20 mg is close to the linearity limit of the balance. However, we conducted several trial measurements with this PM 2000 balance prior to the reported results (see attached additional figures 1-4), and we are confident that our results are valid and reproducible (see also appendix D).

However, in subsequent experiments, we will use a more robust balance to avoid similar challenges.

According to your data, the water uptake by the plant appears to be constant over time. How do you explain the nonconstant change in resistivity over time in light of this observation?

**Reply:** Thanks for your observation. This is due to a combination of two factors:

1. The drying out of the exposed root surface as in the case of desiccation test

2. The uptake of water by the root tip

The baseline measurement was performed with root exposed, then the root tip is placed in water, and measurements are taken at 5 minutes interval. We expect that the surface of the root will dry out first and increase the resistivity, then as the root takes up more water the resistivity becomes more stabilized. This is also supported by the results of replicate measurements (see Appendix D: Figure D1).

It is also challenging to understand how water uptake by the plant is equal or higher than evaporation. Typically, transpiration represents a fraction of the evaporation from a free surface. In this case, the free surface is the water in the tube, and the plant extracts water only from its root tip. Given the relatively small contact area between the root and the water, could you clarify

**Reply:** Thanks for your comment, this experiment was performed in a controlled environment where the temperature and humidity were kept relatively constant, with a focus to reducing evaporation in order to observe and quantify the uptake of water from a single root segment ( see L178-194). We argue that this outcome is actually expected.

Also, we measured evaporation from the tube without the root in it for same duration of the experiment (see Table B1 in appendix), this helped us to properly separate water uptake by the roots from the water loss due to evaporation.

I am concerned about interpreting physiological mechanisms based solely on SIP measurements without supporting data, such as direct measurements of root salt levels. Without corroborating evidence, these claims remain speculative and weaken the study's impact.

**Reply:** Thanks for your comment. While we did not directly measure salt levels in the root, we have evidence that wilting of leaves occurred during the salinity tests in 20 minutes but not during desiccation tests of the same 20 minutes duration (see appendix C)

I recommend reporting such an experiment with repetitions, which is far more important than studying various salt levels at this stage.

**Reply:** For the saline water uptake, we sampled a total of 14 plants [L186-188], thus we argue that the issue of repetitions was accounted for here.

Also, we mentioned earlier that prior to the reported results, we repeated these measurements several times with different replicates of maize and Brachypodium roots and the response to desiccation, water and saline water uptake were all the same [see appendix D: Figure D1, D2 and D3]

**L203:** Could you clarify what you mean by "Maize roots were observed to be more saturated than Brachypodium"? How was the saturation level of the different plants assessed, and what criteria or methods were used to draw this comparison?

**Reply:**  We used the wrong choice of words here, what we mean is that the sampled maize roots were observed to be succulent and white in color, while Brachypodium roots were dry and brownish in color.

We have updated this to read "the sampled maize roots were observed to be succulent and whitish in color, while Brachypodium roots were dry and brownish in color.."

**L212-214:** The statements made here are quite strong. Can you provide supporting evidence or references from the literature to substantiate these claims?

**Reply:** The statements on L212-214 were toned down during the revision to read "Maize roots were probably not plasmolyzed but rather experienced osmotic adjustments".

The following literatures which described "osmotic adjustments" in Maize roots under stress were cited here in the revised manuscript:

1. Sharp et al., 1990: https://doi.org/10.1104/pp.93.4.1337

2. Ogawa and Yamauchi, 2006: https://doi.org/10.1626/pps.9.27

3. Hajlaoui et al., 2010:  https://doi.org/10.1016/j.indcrop.2009.09.007

**Additional Figures**

1. **Desiccation and water uptake**

[Figure]

**Figure D1.** Resistivity and phase spectra of *Brachypodium* (a-b) and maize (c-d) primary roots during demineralized water uptake for 25 minutes. Measurement at 0 minute represents the baseline, measurement was repeated after 5 minutes (to observe drying effect) before putting the root tip in water at 10, 15, 20 and 25 minutes.

**Saline water uptake replicates**

[Figure]

**Figure D2.** Resistivity and phase spectra of *Brachypodium* (a-b) during the uptake of saline water (salt-L) for 25 minutes, and maize (c-d) during saline water (salt-H) uptake for 20 minutes. Measurement at 0 minute represents the baseline, measurement was repeated after 5 minutes (to observe drying effect) before putting the root tip in saline water at 10, 15, 20 and 25 minutes.

[Figure]

**Figure D3.** Resistivity and phase spectra of maize (a-b) during the uptake of saline water (salt-L) for 60 minutes, and (c-d) during saline water (salt-M) uptake for 20 minutes. Measurement at 0 minute represents the baseline, before putting the root tip in saline water.

---

## Author Response (AR2)

**Response to the remaining comments**

**Comment:** While the expanded salt concentration range helps, each salinity level was tested on a single root, limiting statistical robustness. The authors should explicitly acknowledge this limitation in the discussion.

**Reply:** This limitation has been acknowledged in the discussion (see L299-300 of the marked-up manuscript)

**Comment:** The description of the first salt experiment is still unclear – were the same roots sequentially exposed to two salt solutions, or were separate roots used? This should be clarified.

**Reply:** Two roots were used for two different salt solutions, this has been clarified in the text manuscript (see L174-175)

**Comment:** Tables 2 and 3 (water uptake) should indicate whether values represent cumulative or incremental absorption. Some inconsistencies between reported net uptake in the text and table values should be checked

Reply: Table 2 and 3 were already combined into a new Table 2, with only the net uptake reported which agreed with the text (see L222-225)

Also, corrections were made for consistency (see L247)